



# What have we missed when studying the impact of aerosols on surface ozone via changing photolysis rates?

Jinhui Gao[1,2], Ying Li[1], Bin Zhu[3,4], Bo Hu[5], Lili Wang [5], Fangwen Bao[1,2]

[1]Department of Ocean Science and Engineering, Southern University of Science and Technology, Shenzhen, China
[2]School of Earth and Space Sciences, University of Science and Technology of China, Hefei, China
[3]Collaborative Innovation Center on Forecast and Evaluation of Meteorological Disasters, Nanjing University of Information Science and Technology, Nanjing, China
[4]Key Laboratory of Aerosol-Cloud-Precipitation of China Meteorological Administration, Nanjing University of Information Science and Technology, Nanjing, China
[5]State Key Laboratory of Atmosphere Boundary Layer Physics and Atmospheric Chemistry (LAPC), Institute of Atmospheric Physics, Chinese Academy of Sciences, Beijing, China

*Correspondence to*: Ying Li (liy66@sustech.edu.cn)

**Abstract.** Previous studies have emphasized that the decrease in photolysis rate at the surface induced by the light extinction of aerosols could weaken ozone photochemistry and then reduce surface ozone. However, quantitative studies have shown
that weakened photochemistry leads to a much greater reduction in the net chemical production of ozone, which does not match the reduction in surface ozone. This suggested that in addition to photochemistry, some other physical processes related to the variation of ozone should also be considered. In this study, the Weather Research and Forecasting with Chemistry (WRF-Chem) model coupled with the ozone source apportionment method was applied to determine the mechanism of ozone reduction induced by aerosols over Central East China (CEC). Our results showed that weakened ozone
photochemistry led to a significant reduction in ozone net chemical production, which occurred not only at the surface but also within the lowest several hundred meters in the planetary boundary layer (PBL). Meanwhile, a larger ozone gradient was formed in vertical direction, which led to the high concentrations of ozone aloft being entrained by turbulence from the top of the PBL to the surface and partly counteracting the reduction in surface ozone. In addition, ozone in the upper layer of the PBL was also reduced, which was also induced by much ozone aloft being entrained downward. Therefore, by affecting
the photolysis rate, the impact of aerosols was a reduction in ozone not only at the surface but also throughout the entire PBL during the daytime over the CEC in this study. The ozone source apportionment results showed that 41.4%–66.3% of the reduction in surface ozone came from local and adjacent source regions, which suggested that the impact of aerosols on ozone from local and adjacent regions was more significant than that from long-distance regions. The results also suggested that while controlling the concentration of aerosols, simultaneously controlling ozone precursors from local and adjacent
source regions is an effective way to suppress the increase in surface ozone over CEC at present.


## 1 Introduction

Ozone in the troposphere, especially in the planetary boundary layer (PBL), is a well-known secondary air pollutant that is seriously harmful to human health and vegetation (Haagen-Smit and Fox, 1954). As an important source of tropospheric ozone, the photochemical production of ozone is significantly affected by ozone precursors (i.e., $NO_x$ and VOCs) and

photolysis rates, and the latter is determined by the intensity of solar irradiance (Crutzen, 1973; Monks et al., 2015). Aerosols in the troposphere, which are another well-known air pollutant, can influence ozone levels through multiple pathways, for example, modulating temperature (Hansen et al., 1997), light extinction (Dickerson et al., 1997; Gao et al., 2018a), scavenging hydroperoxy ($HO_2$) and $NO_x$ radicals (Li et al., 2019a, b). The light extinction of aerosols can reduce ozone net production (the sum of ozone chemical production and loss) at the surface by reducing the photolysis rate (i.e.,

$J[NO_2]$ and $J[O_3{}^1D]$; Dickerson et al., 1997), which we refer to as the "direct impact". Alternatively, light extinction caused by absorbing aerosols (i.e., black carbon) can suppress the development of the PBL (Ding et al., 2016) and then influence the surface ozone during the daytime (Gao et al., 2018a), which we refer to as the "indirect impact". Studies on the "direct impact" have been conducted in many places around the world (Jacobson, 1998; Castro et al., 2001; Li et al., 2005; Li et al., 2011a), especially in highly polluted regions such as "Beijing-Hebei-Tianjin" region in China (Bian et al., 2007; Deng et al.,

2012; Xing et al., 2017); however, the mechanism of the "direct impact" still has not been fully explained.

Quantitative studies have suggested that, because of the impact of aerosols via affecting photolysis rates, 2%–17% of surface ozone decreased (Jacobson, 1998; Li et al., 2011b; Wang et al., 2016). However, these studies also showed that ozone net production decreased much more (Cai et al., 2013; Wang et al., 2019), which did not match the magnitude of the reduction in surface ozone. For example, a modeling study conducted by Li et al. (2011b) showed that the average reduction in surface

ozone over Central East China (CEC) was -5.4 ppb, whereas the average reduction in ozone net production was -10.5 ppb. The difference between the two reductions indicates that, in addition to ozone photochemistry, there must be other ozone-related physical processes influenced by the reduction in photolysis rate induced by aerosols. However, pertinent studies are still lacking.

At present, air pollution in China is characterized by the "air pollution complex", which shows both aerosols (especially fine

particulate matter $PM_{2.5}$) and ozone pollution issues in the atmosphere (Shao et al., 2006; Li et al., 2017b). With a series of stringent air pollution control policies conducted, the concentrations of aerosols have decreased in the past a few years (Wang et al., 2017); in contrast, the concentrations of ozone in China increased, especially in CEC (Reports on the State of the Environment in China, http://english.mee.gov.cn/Resources/Reports/soe/). Studies have suggested that the extensive reduction in aerosols may cause a potential risk of surface ozone enhancement (Anger et al., 2016; Wang et al., 2016). In this

case, fully understanding and quantifying the impacts of aerosols on ozone is helpful for providing more reasonable advice for air quality protection policies in China.

In this study, the fully coupled "online" model system, Weather Research and Forecasting with Chemistry (WRF-Chem) model, was applied to simulate air pollutants over CEC in October 2018. The impact of aerosols on ozone via influencing the





photolysis rate was quantitatively studied by using process analysis, through a comparison between control and sensitivity
simulations. In addition, with the application of the ozone source apportionment method (Gao et al., 2016; 2017) we
developed and coupled with the WRF-Chem model system, the ozone contributions and their changes induced by aerosols
over typical cities in CEC were also discussed quantitatively in this study. This paper is organized as follows. A description
of the model setting, used data, and scenario design is presented in section 2. The results and discussion of the subject are
presented in section 3. And finally, we end with the conclusions in section 4.

**2 Methodology**

**2.1 Model configuration**

The model system used in this study, the WRF-Chem model, is a fully coupled "online" 3-D Eulerian meteorological and
chemical transport model that has been globally applied in air quality research (Tie et al., 2013; Zhang et al., 2014; Gao et al.,
2018b; Hu et al., 2019). The version of the WRF-Chem model we used in this study is 3.9.1.1, and detailed introductions of
the meteorological parts and chemical parts can be found in Skamarock et al. (2008) and Grell et al. (2005), respectively.
Regarding the simulation settings, two nested domains (Fig. 1) were set up with grid sizes of 122×122 and 150×150 at
horizontal resolutions of 36 km and 12 km for the parent domain (D1) and nested domain (D2), respectively. D1 covered
most parts of China and the surrounding areas and ocean, and D2 covered most parts of East China. The modeling results of
D1 provided meteorological and chemical boundary conditions for the simulations of D2. For the vertical direction, 38 layers
were set up from the surface up to a pressure limit at 50 hPa. It should be noted that 12 layers were located below the lowest
2 km, which is suitable for us to discuss the impacts of aerosols on ozone in the PBL. The Carbon Bond Mechanism Z
(CBM-Z; Zaveri and Peters, 1999) was applied as the gas-phase chemical mechanism in this study. CBM-Z is the upgraded
version of Carbon Bond IV (Gery et al., 1989), which includes 53 species with 133 reactions and extends the framework to
function for larger spatial scale and longer time. Correspondingly, the Model for Simulating Aerosol Interactions and
85  Chemistry with 8 bins (MOSAIC-8bins; Zaveri et al., 2008) was chosen as the aerosol chemistry mechanism. Other
parameterization settings are listed in Table 1.
Since the light extinction of aerosols can impact ozone in two ways, it is necessary to distinguish the direct impact on ozone
in this study. Thus, two parallel experiments were designed in this study: (1) photolysis rate calculation without the presence
of aerosol optical properties (Exp1) and (2) photolysis rate calculation with considering the optical properties of all kinds of
90  aerosols (Exp2). By comparing the results between Exp1 and Exp2, the impact of aerosols on ozone via influencing the
photolysis rate can be determined. Both experiments started at 00:00 UTC on 29 September 2018 and ended at 00:00 UTC
on 31 October 2018. The first two days were designated as the spin-up period.



## 2.2 Description of used data

Many kinds of data were used in this study. The initial and boundary meteorological and chemical conditions were provided by the National Centers for Environmental Prediction (NECP) final (FNL) operational global analysis data and outputs of the Community Atmosphere Model with Chemistry (CAM-chem; Lamarque et al., 2012). Regarding the emissions used in this study, anthropogenic emissions were provided by the Multi-resolution Emission Inventory for China (MEIC; http://www.meicmodel.org/). This inventory includes five anthropogenic sectors (industry, power plant, transportation, residential combustion and agricultural activity), and each section contains both gas and aerosol species ($SO_2$, $NO_x$, $NH_3$, CO, VOCs, BC, OC, $PM_{10}$, and $PM_{2.5}$; Li et al., 2017a). Biogenic emissions were generated by using the Model of Emission of Gas and Aerosols from Nature (MEGAN; Guenther et al., 2006).

Meteorological observations (temperature, wind direction and wind speed) from 110 stations and air pollutants (ozone, $NO_2$ and $PM_{2.5}$) from 110 stations were collected to evaluate the model performance. The locations of the observation stations are presented in Fig. 1b. Hourly meteorological data were measured by the national surface observation network operated by the China Meteorological Administration (CMA). The hourly concentrations of air pollutants were measured and maintained by the China National Environmental Monitoring Center, and published online(http://113.108.142.147:20035/emcpublish). More information on the measurement of air pollutants can be seen in Wang et al. (2014b). In addition, the $NO_2$ photolysis rate ($J$[$NO_2$]) was measured at a comprehensive observation station (116.95°E, 39.75°N; denoted with an up-ward triangle in Fig. 1b). The observation station, attached to the Institute of Atmospheric Physics (IAP) Chinese Academy of Sciences, is located in Xianghe, Hebei Province, approximately 65 km away from Beijing. The photolysis rates were measured by spectroradiometry technique (Hofzumahaus et al., 1999) with a measurement frequency of 10 s and in unit of $s^{-1}$. More information about the measurement technique is available in Hofzumahaus et al. (1999) and Bohn et al. (2004).

## 2.3 Source region settings for ozone source apportionment

Due to secondary pollutant properties, tropospheric ozone is highly dependent on the photochemical reactions of its precursors ($NO_x$ and VOCs). In this study, an ozone source apportionment method was coupled into the WRF-Chem model. This approach, considering both $NO_x$-limited and VOC-limited conditions, is a mass balance technique that identifies the contributions from all geographic source regions to ozone in each grid or region in the model domain within one simulation. This method is similar to the Ozone Source Apportionment Technology (OSAT; Yarwood et al., 1996) which is coupled with the Comprehensive Air quality Model with extensions (CAMx; ENVIRON, 2011), with some modifications to suit the requirements of the WRF-Chem model. More information on the ozone source apportionment method can be found in Gao et al. (2016; 2017).

In this study, 20 geographic source regions were set up in the model domain. The North China Plain and eastern China are two economic hubs in China and suffered serious air pollutions in recent years (Wang et al., 2014a; 2014c; Ding et al., 2016; Kang et al., 2019). As shown in Fig. 1, the two areas are separated into 10 source regions based on administrative divisions.



Other provinces belonging to China and areas outside of China in the model domain are far from CEC but may also
       influence the air quality of CEC under favourable synoptic conditions. Thus, these regions were combined and defined as
       several source regions. Other details of the source regions are listed in Table S1, which can be seen in the supplementary
       material. In addition to the geographic source regions, chemical boundary condition provided by MOZART-4 outputs,
       named $O_{3\text{-Inflow}}$, was defined as an independent contribution, from which the air pollutants can flow into the model domain
and impact ozone in CEC. The initial conditions of D1 (INIT1) and D2 (INIT2) were also settled as independent ozone
       contributions.

### 3 Results and discussion

### 3.1 Model validation

       Although the WRF-Chem model has been widely used in air quality research, the performance varies dramatically when
dealing with different domains, episodes and parameterization settings. In this study, common model performance metrics
       (IOA: Index of Agreement; MB: mean bias; RMSE: root mean square error; MNB: mean normalized bias; MFB: mean
       fractional bias) were used to validate meteorological factors (T2: temperature at 2m above the surface; WS: wind speed at 10
       m above the surface; WD: wind direction at 10 m above the surface) and air pollutants (ozone, $NO_2$ and $PM_{2.5}$). In addition,
       the observed time series of $J[NO_2]$ from Xianghe station was collected and used to validate the model performance for
photolysis rate.

### 3.1.1 Model validation of meteorological and air quality simulations

       For meteorological factors and air pollutants, observation data from more than 100 stations distributed in D2 (Fig. 1b) were
       collected. Considering the large data size, averaged model performance metrics are listed in Table 2. The benchmarks shown
       in brackets follow the recommended values suggested by Emery et al. (2001) and EPA (2005; 2007). In addition, the model
performance of meteorological factors and air pollutants at each station is displayed by the Taylor diagram (Taylor, 2001;
       Gleckler et al., 2008) as shown in Figs. S1 and S2, which are available in the supplementary material.
       Regarding meteorological factors, T2 showed high values of the mean IOA, which was within the scope of its benchmark,
       indicating that the simulation agreed very well with the observations. The mean MB and RMSE of T2 were comparable with
       which in another modeling study (Hu et al., 2016) over the same region and during the same period. However, MB was
slightly beyond the scope of its benchmark, which suggested a slight over-estimation of temperature. Simulations on wind
       speed showed satisfactory model performance since the values of IOA, MB and RMSE all met the criteria. Because of the
       vector nature of wind direction, the IOA of WD followed the calculation recommended by Kwok et al. (2010). The IOA of
       WD reached 0.89, which suggest a good agreement between the simulation and observation on wind direction. In addition,
       the MB was also within the benchmark, which also indicated the satisfactory model performance for wind direction.





For air pollutants, good agreement was found between the simulations and observations since the IOAs of ozone, $NO_2$ and
$PM_{2.5}$ were 0.84, 0.73 and 0.74, respectively. The MNB of ozone was 0.16, which was slightly higher than the benchmark,
while the MFB of $PM_{2.5}$ was within the scope of its benchmark. It should be noted that all of the model performance metrics
of air pollutants were comparable with other modeling studies (Hu et al., 2016; Gao et al., 2018a) over CEC, which also
indicated that our model performance for air pollutants was acceptable.

### 3.1.2 Model validation of $J[NO_2]$


Figure 2a shows the comparison of observed (dark gray dots) and predicted (red line, denotes $J[NO_2]$ in Exp2) $J[NO_2]$ at
Xianghe station. $J[NO_2]$ showed significant diurnal variations due to the strong dependence of photolysis on solar irradiance.
Based on the comparison, the predicted $J[NO_2]$ agreed very well with the observed $J[NO_2]$ and can capture the variation
pattern during the whole Oct. 2018. Comparing the simulated $J[NO_2]$ in Exp1 (blue line in Fig. 2a), the simulated $J[NO_2]$ in
Exp2 agreed better with the observations than that in Exp1 (especially in during "polluted" days with high concentrations of
$PM_{2.5}$), which showed the reasonability of the calculations of the photolysis rate in Exp2 by considering the optical
properties of aerosols. The model performance metrics of Exp2 (presented in the top-right corner of Fig. 2a) also
demonstrate the satisfactory model performance for $J[NO_2]$. High values of IOA (0.99) indicated excellent agreement of the
time series pattern between observations and simulations. MB ($2.0 \times 10^{-4}$) was nearly one order of magnitude smaller than the
average $J[NO_2]$ ($\overline{OBS}$=$1.6 \times 10^{-3}$); in addition, the NMB and NME were also very small, which indicated a small bias between
observations and simulations.

### 3.2 Impact of aerosols on the photolysis rate

As shown in Fig. 2, when the concentrations of $PM_{2.5}$ (Fig. 2b) were low, for example, during the $1^{st}$–$3^{rd}$ and $6^{th}$–$11^{th}$ periods
(the blue shaded parts), the surface $J[NO_2]$ in these two cases were almost the same. However, when examining the polluted
days (the yellow shaded parts), the surface $J[NO_2]$ decreased significantly due to the attenuation of incident solar irradiance
induced by the light extinction of aerosols. It should also be noted that the light extinction of aerosols is not the only factor
that affects the photolysis rate. Clouds can also affect the incident solar irradiance and significantly decrease the photolysis
rate (Wild et al., 2000). That is why $J[NO_2]$ in Exp1 decreased during the daytime on the $15^{th}$ Oct. However, the difference
between Exp1 and Exp2 also reflected the impact of aerosols.

The impact of aerosols on the photolysis rate occurs not only at the surface but also along with the vertical direction. To
investigate the aerosols' impact on the photolysis rate, the $J[NO_2]$ profiles under the low-level aerosol condition (clean) and
high-level aerosol condition (polluted) at noon (12:00) are compared in Fig. 3. The $J[NO_2]$ profiles with surface $PM_{2.5}$
concentrations lower than 35 µg m$^{-3}$ were averaged to represent the $J[NO_2]$ profile under clean conditions (Fig. 3a). The
$J[NO_2]$ profiles with surface $PM_{2.5}$ concentrations greater than 75 µg m$^{-3}$ were averaged to represent the $J[NO_2]$ profile under
polluted conditions (Fig. 3b). The referenced critical values of the surface $PM_{2.5}$ concentration (35 µg m$^{-3}$ and 75 µg m$^{-3}$)
were            determined            based            on            the            national            air            quality            standard





(http://www.cnemc.cn/jcgf/dqhj/201706/W020181008687879597492.pdf). It should be noted that all the selected data was under clear sky conditions, which excludes the impacts of clouds on $J[NO_2]$.

Under clean conditions (Fig. 3a), $PM_{2.5}$ concentrations along with the vertical direction were low (with mean concentrations of 8.6 μg m$^{-3}$ in the PBL and 1.0 μg m$^{-3}$ above the PBL), which suggested that the impact of aerosols on the photolysis rate was small. Consequently, the two profiles did not show significant differences in the vertical direction. Under polluted conditions (Fig. 3b), the concentrations of $PM_{2.5}$ were at a relatively high level in the lowest 1.3 km (with mean value of 90.0 μg m$^{-3}$), especially in the PBL, where the mean concentration of $PM_{2.5}$ reached 123.1 μg m$^{-3}$. In this case, $J[NO_2]$ decreased with height in the lowest 1.3 km, which was due to the attenuation of incident solar irradiance induced by the light extinction of aerosols (Li et al., 2005; Li et al., 2011b). However, at altitude above 1.3 km with lower levels of $PM_{2.5}$, $J[NO_2]$ was enhanced, which could be due to the enhancement of light caused by the light-scattering effect of aerosols (i.e., sulfate aerosols) at the lower height. Our results regarding the changes in the $J[NO_2]$ profile caused by aerosols were consistent with the study of Dickerson et al. (1997).

### 3.3 Impact of aerosols on ozone via decreasing the photolysis rate

### 3.3.1 Changes in surface ozone induced by the decrease of photolysis rate

At the surface, the mean distributions of daytime $PM_{2.5}$ [from 08:00 to 17:00 local time (LT)] under polluted conditions over CEC are presented in Fig. 4a. Correspondingly, the change and relative change in ozone between Exp2 and Exp1 are illustrated in Fig. 4b and 4c, respectively.

High concentrations of $PM_{2.5}$ covered most of the Beijing-Tianjin-Hebei region and the northern Henan Province. In particular, cities with a large population, and large numbers of vehicles and industries, such as Beijing (BJ), Tianjin (TJ), Shijiazhuang (SJZ) and Zhengzhou (ZZ), suffered from more severe particle pollution (mean concentrations were 97.6, 99.8, 113.0 and 79.5 μg m$^{-3}$ in BJ, TJ, SJZ, and ZZ, respectively). The distributions of surface ozone reduction (Fig. 4b and 4c) were similar to the distributions of $PM_{2.5}$ at the surface. More specifically, in the representative cities with severe particle pollution (BJ, TJ, SJZ and TJ), the mean reductions in surface ozone reached 10.6 ppb, 8.6 ppb, 8.2 ppb and 4.2 ppb, respectively, which accounted for 19.0 %, 19.4 %, 17.7 % and 7.9 % of the mean concentrations of surface ozone in these cities, respectively.

Chemical and physical processes analysis (Zhu et al., 2015; Gao et al., 2016) was implemented to discuss the mechanism of the surface ozone reduction induced by aerosols via influencing the photolysis rate in the four representative cities. The following processes were considered: chemistry (CHEM, which is the sum of ozone chemical production and loss of ozone in atmosphere; this contribution is the same as the "ozone net production" which was mentioned in other studies), vertical mixing (VMIX, which is caused by turbulence in the PBL and is closely dependent on turbulence intensity and the vertical gradients of ozone) and advection (ADV, which is caused by the transport effects of wind fields). More information on





processes analysis of the WRF-Chem system is available in Zhang et al. (2014), Gao et al., (2016), and user guide of the WRF-Chem model.

Figure 5 illustrates the mean surface ozone concentrations and processes analysis results of the four cities during 07:00–18:00 (the results of each city are presented in Fig. S3 in the supplementary material). As shown in Fig. 5a, surface ozone began to be reduced by the impact of aerosols starting at 08:00 AM. From then, ozone reduction accumulated until the afternoon, with a maximum value of 11.7 ppb at 14:00. Similar to the processes analysis results of other studies (Kaser et al., 2017; Tang et al., 2017; Xu et al., 2018), the variation in surface ozone was mainly controlled by VMIX and CHEM during 225 the daytime (Fig. 5b and 5c). The contributions of CHEM at the surface were generally below zero, which showed that the chemical consumption of ozone was equal to or stronger than the chemical production of ozone at the surface level. However, the contribution of VMIX was positive, which was the key factor leading to the increase in surface ozone during the daytime. The reduction in surface ozone induced by aerosols can be decomposed into changes in process contributions (Exp2-Exp1), which are shown in Fig. 5d. The contributions of CHEM decreased significantly during the daytime, which was mainly due 230 to the reduction in ozone chemical production caused by weakened ozone photochemistry. Distinct from the change in CHEM, the contribution of VMIX to surface ozone was enhanced during the same period. From 8:00 to 14:00, the reduction in CHEM was more significant than the enhancement in VMIX, which made surface ozone continue decreasing during this period. After 14:00, the enhancement in VMIX almost counteracted the reduction in CHEM, which showed that ozone stopped decreasing. Quantitative results (Table 3) showed the ozone reduction and the accumulated changes in each process 235 contribution at 14:00. The reduction in CHEM was much larger than the reduction in surface ozone. VMIX and ADV were enhanced during this period. The enhancement of ADV was relatively small, whereas the enhancement of VMIX was much stronger, which partly offset the reduction in CHEM. Considering the changes in all of the processes, the change in NET contribution finally equaled to the reduction in ozone. Table 3 clearly illustrates that the offset effect of VMIX led to the inequality between the reduction in CHEM and reduction in surface ozone reported in the study of Li et al (2011b).

**3.3.2 Changes in ozone in the PBL induced by the decrease of photolysis rate**

The averaged vertical changes of processes contributions of the four representative cities are presented in Fig. 6 (the results of each individual city are quite similar and are presented in Figs. S4-S7 in supplementary material). CHEM showed positive contributions aloft in both Exp1 and Exp2 (Fig. 6a and 6e, respectively), which resulted from strong ozone photochemical production. At the surface, it showed negative or weak positive contributions which was attributed to the much stronger 245 chemical loss at the surface caused by ozone consuming species (i.e., NO). Figure 6i shows that the reduction in CHEM induced by aerosols occurred not only at the surface but also within the lowest 500 m during the daytime. VMIX (Fig. 6b and 6f) showed a negative contribution in the upper layer and a positive contribution in the lower layer, which indicated a high concentration of ozone aloft being entrained downward to the surface by turbulence during the daytime (Zhang and Rao, 1999; Gao et al., 2018a). The impact of aerosols enhanced the contributions of VMIX; thus, the change in VMIX showed a 250 positive value within the lowest 300 m and negative values in the upper layer in the PBL. ADV (Fig. 6c and 6g) showed





small contributions, and there was no significant change in ADV caused by the impact of aerosols. NET_DIF reflects the sum of the changes in all of the processes contributions and its distributions showed that, by affecting photolysis rate, the impact of aerosols led to the reduction in ozone occurring not only at the surface but also in the whole PBL (Fig. 6l). In the lower layer of the PBL, the reduction in CHEM was primarily responsible for the reduction in ozone, while the increase in

VMIX partly counteracted the reduction in ozone. In the upper layer of the PBL, the decrease in VMIX played an important role in decreasing ozone aloft.

The contribution of VMIX is closely related to ozone vertical gradients and turbulence exchange coefficients. Studying the changes in the two factors is helpful to investigate the enhancement of VMIX induced by aerosols. As shown in Fig. 7a and 7b, via influencing the photolysis rate, the impact of aerosols didn't cause obvious changes in the exchange coefficients since

the exchange coefficient profiles were almost the same as those from Exp1 and Exp2. However, the ozone gradient from Exp2 was larger than that from Exp1, which suggested that the enhancement of VMIX induced by aerosols was mainly associated with the increase in the ozone gradient. Because of the impact of aerosols, the chemical reduction in ozone was more significant in the lower layer than in upper layer in the PBL (Fig. 6i), which led to smaller concentrations of ozone in the lower layer and consequently formed a larger vertical gradient (Fig. 7c). Therefore, high concentrations of ozone aloft

would be entrained from the top of the PBL to the surface, which leaded to the enhancement in VMIX. In addition, similar features also occurred in each representative city which can be seen in Fig. S8 in the supplementary material.

### 3.4 The changes in ozone source contributions induced by aerosols via influencing the photolysis rate

Figure 8 illustrates the average ozone contributions from geographic source regions to surface ozone in the four cities from Exp1 and Exp2, and the changes in each ozone contribution induced by aerosols are also presented. For the representative

cities, surface ozone was mainly contributed by local contribution and the contributions from adjacent source regions (left and middle columns in Fig. 8). For example, surface ozone over BJ and TJ was mainly contributed by ozone from themselves and Hebei Province. For SJZ and ZZ, ozone from their respective provinces (HB and HN) contributed more significantly than ozone from other regions did. In addition, $O_{3\text{-Inflow}}$, which can be approximately treated as background ozone (Gao et al., 2017), also showed an obvious contribution to surface ozone over each city.

With the impacts of aerosols, ozone from local and adjacent source region decreased more significantly than ozone from further source region did (right column in Fig. 8). For each city, the first four source regions that ozone contribution changed the most to the mean ozone concentration from 13:00 to 16:00 are listed in Table 4. For BJ and TJ, which are defined as independent source regions, ozone from local region decreased by -3.8 ppb and -3.8 ppb to BJ and TJ, respectively, which accounted for the greatest proportion. In addition, HB is adjacent to BJ and TJ, and ozone from HB decreased by 3.1 ppb and

3.0 ppb to ozone in BJ and TJ, respectively, which was more than ozone from long distance source regions did. SJZ and ZZ are the provincial capitals of HB and HN, ozone from HB and HN decreased by 4.6 ppb and 5.8 ppb to SJZ and ZZ, respectively. The reduction in ozone at the surface was mainly caused by the reduction in chemical production. For the ozone source apportionment method in this study, ozone chemical production can be traced to the source based on the ratio





of ozone precursors from each source region. Due to the short lifetime of ozone precursors (i.e., $NO_x$), there will be more
ozone precursors from local and adjacent source regions than which from further source regions. Thus, surface ozone from
local and adjacent source regions decreased more with the impact of aerosols. At present, surface ozone has increased
annually since the reduction in aerosols. Our ozone source apportionment results suggest that controlling ozone precursors
from local and adjacent regions will be a more effective way to suppress the increase in surface ozone over CEC.

## 4 Conclusions

Currently, in China, the concentrations of surface ozone increase annually, which is considered closely related to the
decrease in $PM_{2.5}$. Previous studies have summarized that, by decreasing the photolysis rate at the surface, the light
extinction of aerosols could weaken ozone photochemistry and then directly reduce surface ozone. However, quantitative
studies showed that the reduction in ozone net chemical production was much greater than the reduction in surface ozone,
which suggested that some other physical processes related to the variation in surface ozone were not discussed in previous
studies.

To more clearly understand the impact of aerosols on ozone via affecting the photolysis rate, the WRF-Chem model was
applied to simulate air pollutants over CEC in October 2018. Comprehensive model validations demonstrated the model
performance in simulating air quality over CEC during this period. By comparing the results between the control and
sensitive simulation, the mechanism of the impacts of aerosols on ozone was quantitatively studied. With the application of
the ozone source apportionment method that we coupled into the WRF-Chem model, the impact of aerosol on the source-
receptor relationship of ozone was also discussed.

Our results showed that, because of the light extinction of aerosols, the attenuation of incident solar irradiance caused the
decrease in the photolysis rate below the PBL and then weakened ozone photochemistry. In this case, the net chemical
production of ozone was significantly decreased within the lowest several hundred meters in the PBL. The significant
reduction in the net chemical production formed a larger ozone vertical gradient. And more air mass aloft with high
concentration of ozone was entrained downward from the top of the PBL to the surface, which partly counteracted the
reduction in ozone net chemical production. Changes in the two processes together led to the reduction in surface ozone. In
addition, ozone in the upper layer of the PBL was also reduced, which was also induced by much ozone aloft being entrained
downward. Therefore, by affecting the photolysis rate, the impact of aerosols can reduce ozone not only at the surface but
also in the entire PBL during the daytime over CEC in this study.

The ozone source apportionment results showed that, for the four representative cities in CEC (BJ, TJ, SJZ, and ZZ), ozone
from local and adjacent regions decreased by 6.9 ppb, 6.8 ppb, 4.6 ppb, and 5.8 ppb, respectively, which accounted for
41.4%–66.3% of the reduction in surface ozone in these cities. This suggested that the impact of aerosols on ozone from
local and adjacent regions is more significant than that from long-distance regions. In recent years, with the implementation
of the toughest-ever clean air policy in China, aerosols have decreased, whereas ozone increases year by year. Our results





suggest that while controlling the concentrations of aerosols, controlling ozone precursors from local and adjacent regions is an effective way to suppress the increase in surface ozone.

*Acknowledgements.* This work was mainly supported by grants from the National Key Research and Development Program of China (2016YFA0602003), National Natural Science Foundation of China (41905114, 41961160728, 41575106), Science and Technology Planning Project of Guangdong Province of China (Grant 2017A050506003), Shenzhen Peacock Teams Plan (KQTD20180411143441009). Part of this work was supported by China Postdoctoral Science Foundation (2019M662169, 2019M662199). We also want to thank for the support from SUSTC Presidential Postdoctoral Fellowship.

The simulated results in this study were calculated using computational resources provided by the Southern University of Science and Technology. All the observations and model outputs mentioned in this study are available by contacting Ying Li via liy66@sustech.edu.cn.

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



**Table 1: Major configuration options of WRF-Chem used for this study.**

| Item | Selection | Reference |
|---|---|---|
| Photolysis scheme | Fast-J photolysis | Wild et al., (2000) |
| Long wave scheme | RRTMG[a] | Iacono et al., (2008) |
| Short wave scheme | RRTMG[a] | Iacono et al., (2008) |
| Microphysics scheme | Lin scheme | Lin et al., (1983) |
| Land surface scheme | Noah land surface model | Chen and Dudhia (2001) |
| PBL scheme | Yonsei University (YSU) scheme | Hong et al., (2006) |
| Dry deposition scheme | Wesely scheme | Wesely (1989) |

[a] RRTMG=Rapid Radiative Transfer Model for GCMs



**Table 2: Mean model performance metrics for meteorological factors and air pollutants. The values that do not meet the benchmarks are denoted in bold.**

| Variables | IOA | MB | RMSE | MNB | MFB |
|---|---|---|---|---|---|
| T2 | 0.93 (≥0.8) | **0.71** ([-0.5,0.5]) | 2.42 | -0.01 | -0.09 |
| WS | 0.78 (≥0.6) | -0.42 ([-0.5,0.5]) | 1.26 (≤2) | -0.03 | -0.28 |
| WD | 0.89 | 6.59 ([-10,10]) | -0.42 | 1.64 | 0.02 |
| O$_3$ | 0.84 | -6.51 | 27.68 | **0.16** ([-0.15,0.15]) | -0.24 |
| NO$_2$ | 0.73 | -5.97 | 23.39 | -0.13 | -0.35 |
| PM$_{2.5}$ | 0.74 | 8.11 | 28.75 | 0.34 | 0.08 ([-0.6,0.6]) |




**Table 3: The reduction of surface ozone at 14:00 and the corresponding accumulated changes of processes contributions.**

| $\Delta O_3^{at\ 14:00}$ | $\sum_{i=8:00}^{14:00} CHEM\_DIF_i$ | $\sum_{i=8:00}^{14:00} VMIX\_DIF_i$ | $\sum_{i=8:00}^{14:00} ADV\_DIF_i$ | $\sum_{i=8:00}^{14:00} NET\_DIF_i$ |
|---|---|---|---|---|
| -11.7 ppb | -44.2 ppb | 31.6 ppb | 0.9 ppb | -11.7 ppb |






**Table 4: The first four source regions that ozone contribution changes the most to the mean ozone concentration from 13:00 to 16:00 in each city. Local region and source region where the city located in are denoted as bold.**

| City | ΔOzone | ΔContribution | | | |
| --- | --- | --- | --- | --- | --- |
| | | 1st | 2nd | 3rd | 4th |
| BJ | -10.4 ppb | **BJ** | HB | TJ | SD |
| | | -3.8 ppb | -3.1 ppb | -1.3 ppb | -0.5 ppb |
| | | (36.5%) | (29.8%) | (12.5%) | (4.8%) |
| TJ | -12.3 ppb | **TJ** | HB | SD | SIB |
| | | -3.8 ppb | -3.0 ppb | -1.9 ppb | -0.8 ppb |
| | | (30.9%) | (24.4%) | (15.4%) | (6.5%) |
| SJZ | -11.1 ppb | **HB** | HN | SIB | $O_{3\text{-inflow}}$ |
| | | -4.6 ppb | -1.5 ppb | -0.9 ppb | -0.8 ppb |
| | | (41.4%) | (13.5%) | (8.1%) | (7.2%) |
| ZZ | -9.8 ppb | **HN** | JS | SIB | SH |
| | | -5.8 ppb | -0.9 ppb | -0.6 ppb | -0.4 ppb |
| | | (59.2%) | (9.2%) | (6.1%) | (4.1%) |






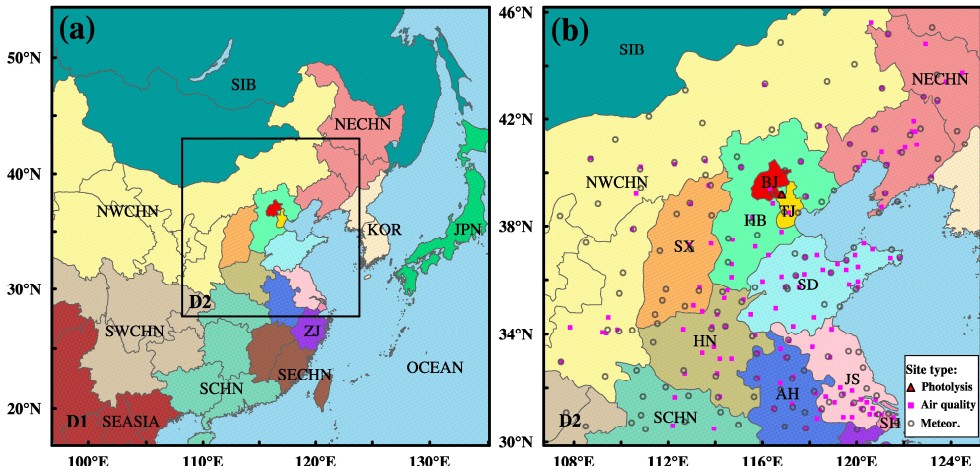

**Figure 1: Model domain. Hundreds of observations are used for model validation, locations and types of the observation stations are shown in (b). The figure also shows the source regions denoted by different colors.**





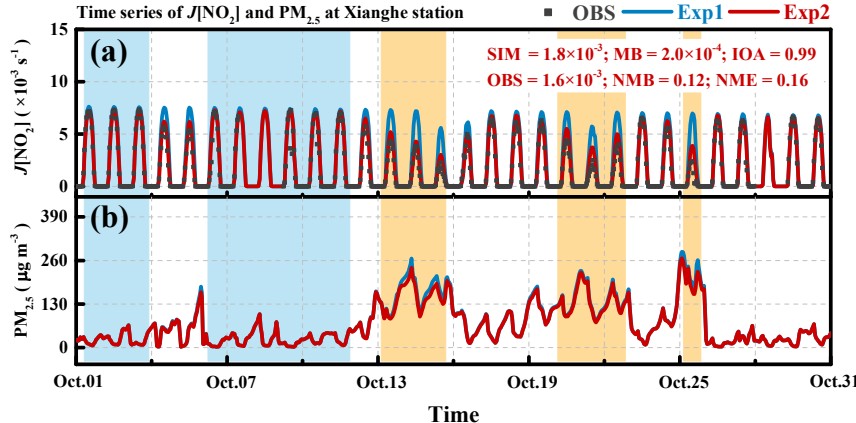


**Figure 2: Time series of simulated $J[NO_2]$ (a) and $PM_{2.5}$ (b) at Xianghe station.**

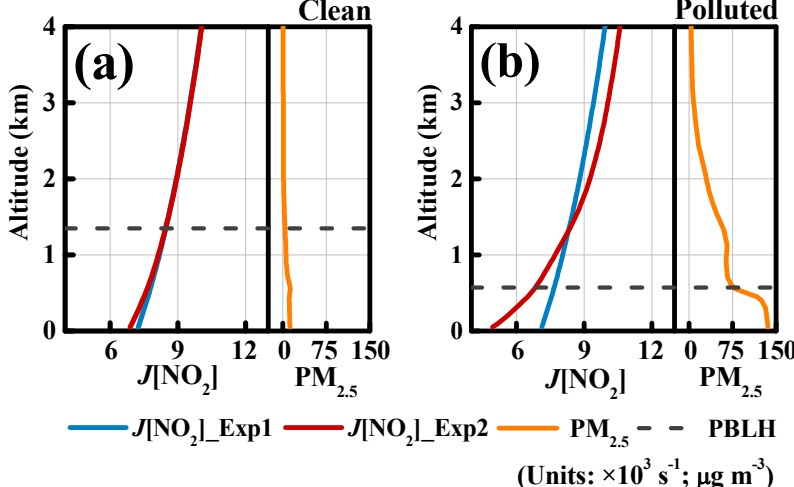

**Figure 3:** Mean profiles of $J[NO_2]$ (red and blue lines) and $PM_{2.5}$ (orange line) at 12:00 in clean days (a) and polluted days (b). Profiles of $J[NO_2]$ in Exp1 and Exp2 are denoted by red and blue, respectively. Mean PBL heights (PBLH; black dashed line) of the two kinds of days are also presented (a) and (b), respectively.




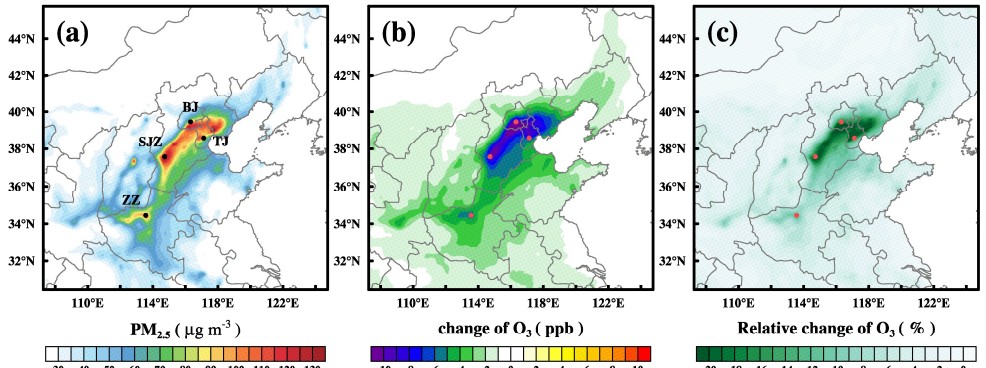

**Figure 4: Mean distributions of PM$_{2.5}$ (a), change of O$_3$ (b) and relative change of O$_3$ (c) at surface over CEC during high PM$_{2.5}$**
**episodes. Dots denote the four typical cities in CEC, BJ=Beijing; TJ=Tianjin; SJZ=Shijiahuang; ZZ=Zhengzhou**





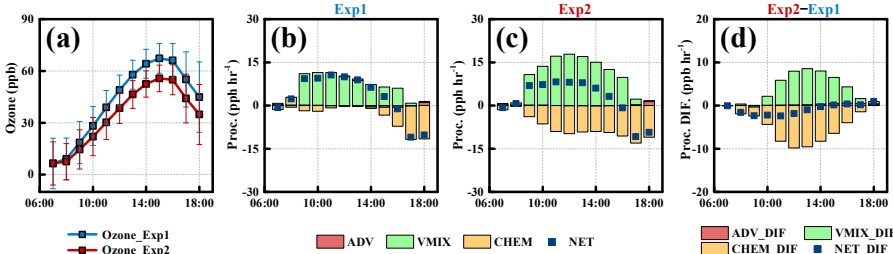

**Figure 5: Averaged ozone concentrations at surface (a), hourly processes contributions and net contribution of ozone in Exp1 (b) and Exp2 (c), the changes of hourly processes contributions and net contribution induced by aerosol (Exp2-Exp1) at daytime. CHEM = chemistry, VMIX = vertical mixing, ADV = advection, NET = CHEM + VMIX + ADV. Changes of each process contribution and NET contribution are denoted by CHEM_DIF, VMIX_DIF, ADV_DIF and NET_DIF, respectively.**



**Exp1**

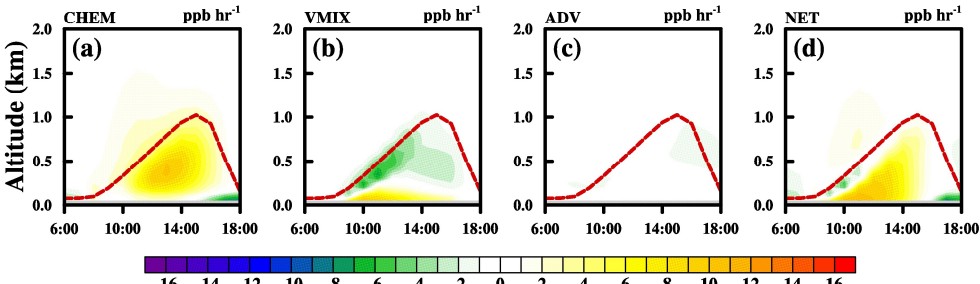

**Exp2**

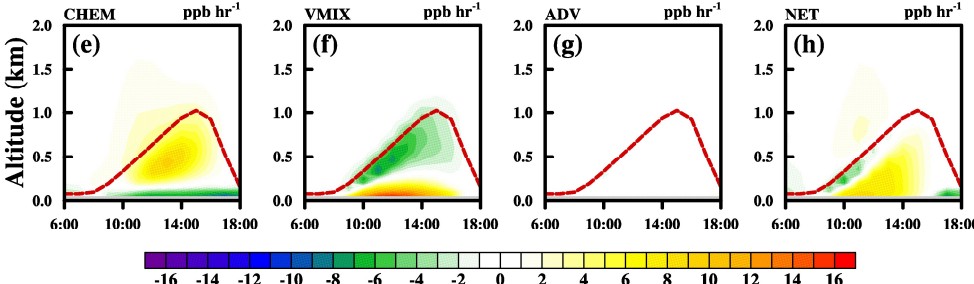

**Exp2-Exp1**

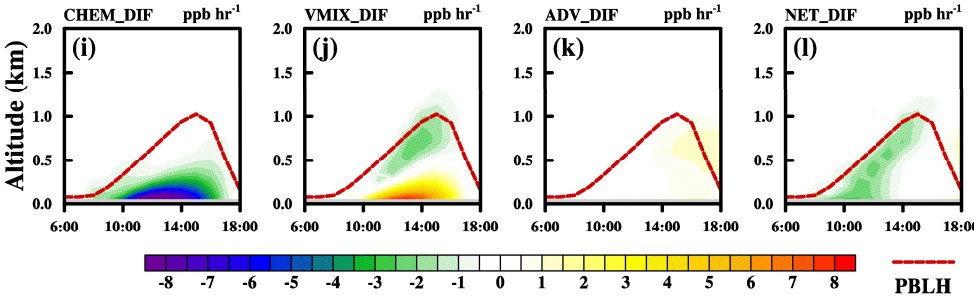

**Figure 6: Averaged vertical distributions of processes contributions in function with time from 06:00 to 18:00 LT. (a)-(d) for CASE1; (e)-(h) for CASE2; (i)-(l) for the changes of each process contribution due to aerosols (Exp2-Exp1). Red dash lines denote PBLH.**




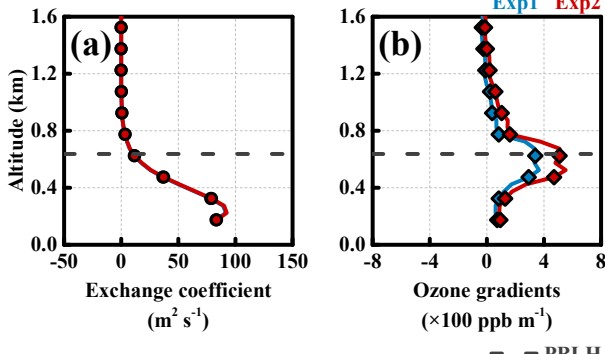

**Figure 7:** Averaged vertical profiles of (a) turbulence exchange coefficients and (b) vertical gradients of ozone from Exp1 and Exp2 at 12:00 AM. Dark gray dash line denotes PBLH at this time.

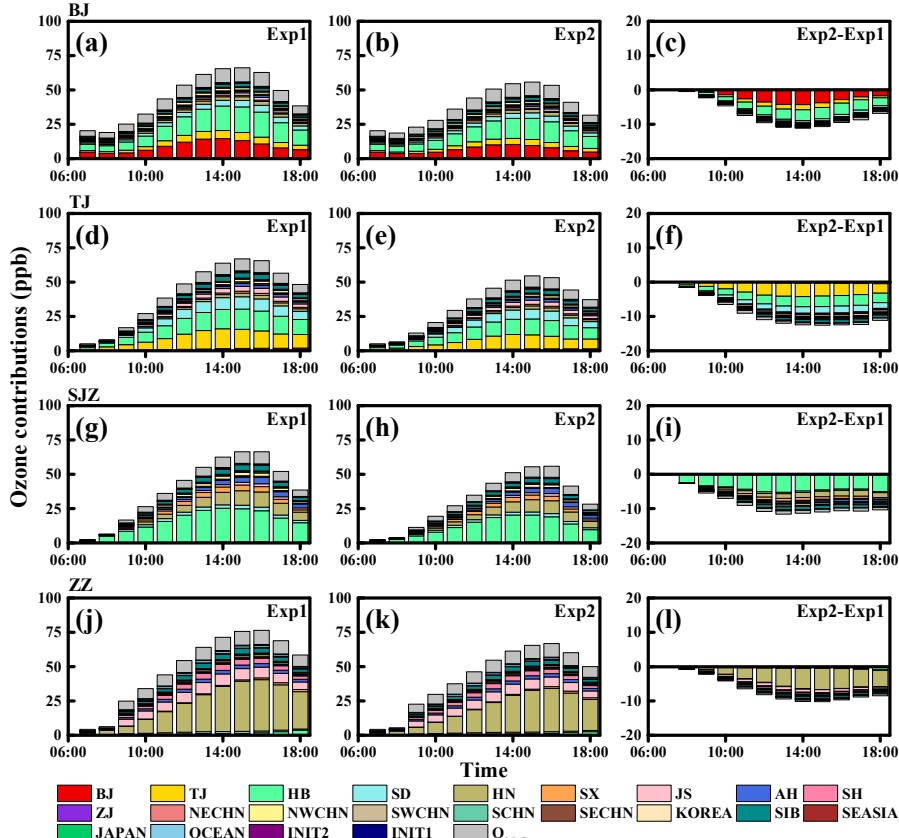

**Figure 8: Averaged ozone contributions and changes induced by aerosols from geographical source regions to BJ (a)-(c), TJ (d)-(f), SJZ (g)-(i) and ZZ (j)-(l) from 07:00 to 18:00 LT.**
