# Peer review of "What have we missed when studying the impact of aerosols on surface ozone via changing photolysis rates?"

_Atmospheric Chemistry and Physics, 2020_

## Referee Comment (RC1) · Anonymous Referee #2 · 15 Apr 2020

**Review for Gao et al. 2020 submitted to ACPD.**

**General comments**

In this manuscript, the authors focus on the radiative effect of aerosols on the tropospheric ozone photochemistry over China. Specifically, they address the discrepancy (reported in the previous studies) of the substantial decrease of the net ozone production due to aerosols and a much smaller magnitude of the actual ozone decrease at the surface. The authors highlight that the missing mechanism is the vertical entrainment, which enhances due to increased vertical gradients of the ozone in the PBL. They also address the possible mitigation of the recent increase in the surface ozone levels due to lower levels of the ambient PM concentrations. The apportionment method revealed that the local and adjacent regions play the dominant role.

The results of this work are based on the WRF-Chem simulations, ozone tagging technique and extensive observation. The analysis is mostly sound, the manuscript is well written, but some details are missing. I recommend a minor revision, which should give authors enough time to address the comments listed below.

**Specific comments**

I have one major concern regarding the mass balance technique, its description, and the interpretation of the results. Any balance equation inevitably has the residual term, which includes "other" processes (uncategorized) and the numerical error term. During the analysis, one has to show that the specified terms (for example, net chemical production or vertical entrainment) are much larger than this residual term.

Could authors, please, add the mass balance equation and the short description (possibly to the supplementary), which explicitly states all of the terms and address the following questions:
1. How well does the mass balance equation balance? What is the magnitude of the "numerical error" term compared to the other terms? Often, this term has the same order of magnitude. In this study, however, the agreement is exact (for example, Tables 3 and 4).
2. Was the dry deposition taken into account? If it is incorporated into one of the three considered terms (CHEM, ADV, or VMIX), then it has to be extracted and presented separately, or the terms should be renamed (for example, CHEM+DEP).
3. How was the NET term obtained (for example, in figure 5)? Is it merely a sum of ADV+VMIX+CHEM terms or is obtained directly from the separate WRF-Chem output variable, which represents the d[X]/dt?

Given the reasonable model-observation comparison statistics, net chemical production (CHEM) and vertical mixing (VMIX) are likely the only major drivers, but the scientific analysis has to be rigorous.

My minor concern is related to the effect of aerosols on PBL height. In section 3.2 and Figure 3, the authors contrast the clean and polluted cases. The impact of the aerosols on PBL height is vivid (Figure 3, polluted case). However, no discussion on the aerosols properties or the effect of the collapsed PBL is offered. I think that manuscript will improve if authors describe in the text the composition of the aerosols (primary type) and the absorption properties (single scattering albedo). Additionally, considering that tracers are well mixed within the PBL, the PBL reduction by a factor of 2 translates roughly in a two-times increase in tracer concentrations. What role does the PBL collapse play in the adjustment of the surface ozone concentration compared to the aerosols effect on photolysis and vertical entrainment?

**Technical corrections**

1. References should be formatted appropriately and numbered.
2. Please, add units to Table 2.
3. Line 224. Delete "which showed that ozone stopped decreasing"
4. Figure 6, please, update the caption, remove the CASE* and explain that data was spatially sampled and represent four cities.

I enjoyed reading the manuscript and would like to thank authors for the accurate choice of the diagnostics and figures.

---

## Referee Comment (RC2) · Anonymous Referee #1 · 2 May 2020

In this manuscript, the authors examined the effects of aerosols on surface ozone via affecting the photolysis rates over CEC China. They focused on the discrepancy between the decrease of the ozone net production and the smaller magnitude of the ozone reduction at the surface, which have been reported in previous studies. The modeling results suggest that weakened ozone photochemistry led to a significant reduction in ozone net chemical production, which occurred not only at the surface but also within the lowest several hundred meters in the planetary boundary layer. Meanwhile, the authors highlighted that vertical mixing played an important role and partly counteracting the significant decrease in ozone net production at the surface. The ozone apportionment results highlighted more significant impacts on ozone from local

[Figure]

**[ACPD](https://www.atmospheric-chemistry-and-physics.net/)**

Interactive
comment

and adjacent regions, suggesting that while controlling the concentration of aerosols, simultaneously controlling ozone precursors from local and adjacent source regions is an effective way to suppress the increase in surface ozone over CEC at present.

The figures in the manuscript and supplement information are with high quality and well presented. The analysis is sound, but some details need to be revised before the manuscript being published.

The comments are listed below:

1ãĂĄ In model validation (section 3.1), the authors compare the simulated J[NO2] with the observations. In addition to J[NO2], J[O31D] is also important in affecting the ozone photochemical production. Comparison on J[O31D] will show more sufficient evidence to demonstrating the well model performance in simulating photolysis rates. If the authors have the observations of J[O31D], please add the comparison of J[O31D].

2ãĂĄ The authors showed that J[NO2] was enhanced at altitude above 1.3km which is due to the enhancement of light caused by the light-scattering effect of aerosols. Discussions on the compositions of the aerosols and their effects on J[NO2] over this place are necessary. Please add them in the manuscript.

Technical correction: 1ãĂĄ Line 99, add a comma after "combustion" 2ãĂĄ Line 135, add a comma after "episodes" 3ãĂĄ Variables in Table2 need to be added with units. 4ãĂĄ Caption of figure 6 needs to be update. "CASE1" and "CASE2" should be replaced by "Exp1" and "Exp2".
* * *

---

## Author Comment (AC1) · 15 Jul 2020

Dear editor and referee#2,

Thank you very much for your time and attentions on this work. The comments and suggestions are very useful to improve our manuscript. Following is a point-by-point response to referee #2's comments. Texts in black are the comments, those in blue are our responses. All the line numbers mentioned in responses are referred to the manuscript with changes marked.

We hope that you will find the changes satisfactory and we are looking forward to hearing from you soon.
* * *
I have one major concern regarding the mass balance technique, its description, and the interpretation of the results. Any balance equation inevitably has the residual term, which includes "other" processes (uncategorized) and the numerical error term. During the analysis, one has to show that the specified term (for example, net chemical production or vertical entrainment) are much larger than this residual term.
Could authors, please, add the mass balance equation and the short description (possibly to the supplementary), which explicitly states all of the terms and address the following questions:

(1) How well does the mass balance equation balance? What is the magnitude of the "numerical error" term compared to the other terms? Often, this term has the same order of magnitude. In this study, however, the agreement is exact (for example, Table 3 and 4)

Reply: Thank you for your comment. In WRF-Chem model, chemical species undergo the physical and chemical processes that are presented in Fig. R1. The contribution of each process at each time step is represented by the difference between the concentration after the process calculation ($C^{new}$) and the concentration before the

process calculation ($C^{old}$).

[Figure]

Figure R1: Schematic showing the calculation flow of chemical species in the WRF-Chem model.

In this case, for any chemical species at any grid, the change of concentration ($\Delta C$) at each time step equals to:

$$\Delta C = ADV + EMISS + VMIX + DRY + CONV + CHEM + CLOUD + AERO \\ + WET$$

Among them, ADV is the contribution from advection; EMISS is the contribution from direct emission; VMIX is the contribution from vertical mixing process; DRY is the contribution from dry deposition; CONV is the contribution from convection; CHEM is the contribution from gas-phase chemistry; CLOUD is the contribution from cloud chemistry; AERO is the contribution from aerosol chemistry; WET is the contribution

from wet deposition. For ozone, some of the terms don't contribute the change of ozone (ΔOzone). As a typical secondary pollution, there is no direct emission of ozone in the model and EMISS is 0.0. MOSAIC was used as aerosol chemistry mechanism in this study. This mechanism involves processes such as heterogeneous reactions, gas-particle mass transfer process, nucleation and coagulation (Zaveri et al., 2008). However, ozone doesn't participate in the relevant calculations which means AERO is 0.0. Same as aerosol chemistry, ozone also doesn't participate in the calculation of cloud chemistry and the CLOUD is 0.0. Because we selected the simulated data which was under the clear sky conditions, there was no contribution from wet deposition (WET=0.0). In this case, for ΔOzone, the mass balance equation can be simplified to:

$$\Delta Ozone = ADV + VMIX + DRY + CHEM + CONV$$

Since occurring on the ground level, DRY only shows contribution in the first layer in the model domain. Thus, the mass balance equation of ΔOzone at any grid and at each time step can be shown as:

$$\Delta Ozone = \begin{cases} ADV + CHEM + VMIX + CONV + DRY & in\ 1^{st}\ layer \\ ADV + CHEM + VMIX + CONV & above\ 1^{st}\ layer \end{cases}$$

The original WRF-Chem model has provided some processes diagnostic variables (names of these variables are advz_o3, advh_o3, chem_o3, vmix_o3, conv_o3) to show the contributions of the primary processes to ozone concentrations. According to the original model code, each process diagnostic variable is the accumulation of the difference of the ozone concentration between after and before the corresponding process calculation at each time step. The variables advh_o3 and advz_o3 represent the contributions from horizontal advection and zonal advection. And the contribution of ADV is the sum of advh_o3 and advz_o3 (ADV=advh_o3+advz_o3). The variable chem_o3 represents the contribution of CHEM. Dry deposition occurred at surface which is located in the first layer in the model domain. It is calculated together with

vertical mixing process in the subroutine vertmx (chem/module_vertmx_wrf.F) which belongs to the module of dry_dep_driver (chem/dry_dep_driver.F). Thus, in the first layer, the variable vmix_o3 is the sum contribution of VMIX and DRY. And above the first layer, vmix_o3 equals to the contribution of VMIX. In order to discussing processes contributions on surface ozone more clearly, the contributions of DRY and VMIX have been separated from the vmix_o3 (method can be check in the reply of question 2 and the supplementary material) and we ran the two experiments again. In addition, conv_o3 represents the contribution of CONV which may impact ozone concentration when it occurred in the atmosphere. However, in this study, contribution of CONV was 0.0 during the periods we discussed and was not mentioned in our manuscript. The description on mass balance for WRF-Chem model has been presented in the supplementary material.

From the above, it's clear that ΔOzone at any grid and at any time step equal to the sum of the processes contributions mentioned in the manuscript and it also suggested that the mass balance of our simulated results kept well. Because of the calculation method of the processes contributions, the numerical error between the change of ozone concentration and processes contributions is relatively small. However, there are still some numerical error caused by the calculation accuracy in our results. Taking the results in Table 3 in manuscript as an example (shown in Table R1):

Table R1. Detail information for Table 3 in the manuscript.

| $\Delta O_3^{at}$ 14:00 | $\sum_{i=8:00}^{14:00} CHEM\_DIF_i$ | $\sum_{i=8:00}^{14:00} VMIX\_DIF_i$ | $\sum_{i=8:00}^{14:00} DRY\_DIF_i$ | $\sum_{i=8:00}^{14:00} ADV\_DIF_i$ | $\sum_{i=8:00}^{14:00} NET\_DIF_i$ |
|---|---|---|---|---|---|
| -11.70516 ppb | -44.28622 ppb | 12.00781 ppb | 19.58756 ppb | 0.91944 ppb | -11.77141 ppb |

As shown in Table R1, at 14:00, the difference of ozone between Exp1 and Exp2 (Exp2-Exp1) is -11.70516 ppb. The accumulated tendency of the change in each process between Exp1 and Exp2 from 08:00 to 14:00 is also listed in Table R1. $\sum_{08:00}^{14:00} CHEM\_DIF$, $\sum_{08:00}^{14:00} VMIX\_DIF$, $\sum_{08:00}^{14:00} DRY\_DIF$, $\sum_{08:00}^{14:00} ADV\_DIF$ and are -44.28622 ppb, 12.00781 ppb, 19.58756 ppb, and 0.91944 ppb, respectively. And their

sum ($\sum_{08:00}^{14:00} NET\_DIF$) is -11.77141 ppb.

$$\sum_{08:00}^{14:00} NET\_DIF = \sum_{08:00}^{14:00} ADV\_DIF + \sum_{08:00}^{14:00} VMIX\_DIF + \sum_{08:00}^{14:00} CHEM\_DIF + \sum_{08:00}^{14:00} DRY\_DIF$$

We can see that the bias between $\Delta O_3^{at\ 14:00}$ and $\sum_{08:00}^{14:00} NET\_DIF$ is 0.06625 ppb which is much less than other terms. In order to making the table clear, all data in table 3 reserved a decimal fraction and the bias became 0.1 ppb.

(2) Was the dry deposition taken into account? If it is incorporated into one of the three considered terms (CHEM, ADV, or VMIX), then it has to be extracted and presented separately, or the terms should be renamed (for example, CHEM+DEP)

Reply: Thank you for your comment. Dry deposition was taken into account. In previous version of this manuscript, we used result of vmix_o3 to represent the contribution of VMIX. Since the dry deposition and vertical mixing being calculated together in WRF-Chem model, vmix_o3 in the first layer actually contained the contributions of DRY and VMIX. Thus, the VMIX mentioned in section 3.3.1 in previous version of the manuscript contained the contribution of DRY. In order to making the discussion of process analysis on surface ozone more clearly, we followed the comment and separated the contributions of DRY and VMIX from vmix_o3.

It has been known that, pressure and temperature are not changed when doing the dry deposition calculation. Thus, the contribution of DRY to ozone ($C_{O3}$) at each time step (dt) can be calculated as:

$$DRY = C_{O3} * dvel * dt/dz$$

In which, dvel is the dry deposition velocity of ozone and dz is the height of the grid. And the contribution of VMIX in the first layer at each time step equals to:

$$VMIX = vmix\_o3 - DRY$$

Relevant modifications of the code were added into WRF-Chem model. And we ran the two experiments again. The results showed that, since the decrease of surface ozone, the dry deposition of ozone was weakened which leading to the change in DRY

increased during daytime. In addition, the change in VMIX was increased which is due to the enhancement of the vertical mixing process. The increases in DRY and VMIX partly counteracted the reduction in CHEM. Relevant discussion in section 3.3.1, Fig. 5, and Table 3 have been revised, please check the details in the revised manuscript.

(3) How was the NET term obtained (for example, in figure 5)? Is it merely a sum of ADV+VMIX+CHEM terms or is obtained directly from the separate WRF-Chem output variable, which represented the d[X]/dt?

Reply: The NET contribution is the sum of all the processes contributions.

For any grid in the first layer:

$$NET = ADV + CHEM + DRY + VMIX$$

For any grid above the first layer:

$$NET = ADV + CHEM + VMIX$$

In the manuscript, we used NET to represent the hourly net contribution from all the mentioned processes.

(4) Given the reasonable model-observation comparison statistics, net chemical production (CHEM) and vertical mixing (VMIX) are likely the only major drivers, but the scientific analysis has to be rigorous.

My minor concern is related to the effect of aerosols on PBL height. In section 3.2 and Figure 3, the authors contrast the clean and polluted cases. The impact of the aerosols on PBL height is vivid (Figure 3, polluted case). However, no discussion on the aerosols properties or the effect of the collapsed PBL is offered. I think that manuscript will improve if authors describe in the text the composition of the aerosols (primary type) and the absorption properties (single scattering albedo). Additional, considering that tracers are well mixed within the PBL, the PBL reduction by a factor of 2 translates roughly in a two-times increase in tracer concentrations. What role does the PBL collapse play in the adjustment of the surface ozone concentration compared to the aerosols effect on photolysis and vertical entrainment?

Reply: Thank you for your comment. Based on the optical properties of aerosols, they

can be classified into light-scattering aerosols and light-absorbing aerosols. Before talking about the comprehensive effects of aerosols on $J[NO_2]$, it's necessary to present the effects of light-scattering aerosols and light-absorbing aerosols on $J[NO_2]$, respectively.

[Figure]

Figure R2. Time series (a) and mean contributions (b) of the simulated aerosol species at Xianghe station during Oct. 2018. I for the whole month; II for clean days (blue shaded parts in a); III for polluted days (yellow shaded parts in a).

In this study, MOSAIC-8bins was used as the aerosol chemistry mechanism. This mechanism includes eight aerosols species: Sulfate ($SO_4$), Nitrate ($NO_3$), Ammonium ($NH_4$), Sodium (Na), Chlorine (Cl), Organic Carbon (OC), Black Carbon (BC), and, Other Inorganics (OIN). Based on Fig. 2c in manuscript, concentrations of all the simulated aerosols species and their relative contributions to the total concentration of $PM_{2.5}$ at Xianghe station are shown in Fig. R2. During Oct. 2018, the mean concentration of $PM_{2.5}$ was 68.0 µg m$^{-3}$ at Xianghe station. Among all the species, $NO_3$ and OIN contributed significantly which accounted for 30% and 28% to the total concentration of $PM_{2.5}$; $SO_4$, $NH_4$, BC, and OC accounted for ~10%, respectively; Na and Cl showed few contributions during Oct. 2018. Under the "clean" condition (blue shaded parts in Fig. R2a and the pie chart II in Fig. R2b), the mean concentration of

PM$_{2.5}$ decreased to 25.3 μg m$^{-3}$ and OIN contributed (accounted for 38%) more than NO$_3$ did (accounted for 10%). On the contrary, OIN contributed (accounted for 24%) less than NO$_3$ did (accounted for 38%) when it was under the "polluted" condition (yellow shaded parts in Fig. R2a and the pie chart III in Fig. R2b).

Table R2. Refractive indexes of the aerosol species at each wave band in WRF-Chem model

| wave band | 300nm | | 400nm | | 600nm | | 999nm | |
|---|---|---|---|---|---|---|---|---|
| refr. index[a] species | real[b] | imaginary[c] | real | imaginary | real | imaginary | real | imaginary |
| SO4 | 1.52 | $1.00\times10^{-9}$ | 1.52 | $1.00\times10^{-9}$ | 1.52 | $1.00\times10^{-9}$ | 1.52 | $1.75\times10^{-9}$ |
| NO3 | 1.50 | 0.00 | 1.50 | 0.00 | 1.50 | 0.00 | 1.50 | 0.00 |
| NH4 | 1.50 | 0.00 | 1.50 | 0.00 | 1.50 | 0.00 | 1.50 | 0.00 |
| Na | 1.51 | $8.66\times10^{-7}$ | 1.50 | $7.02\times10^{-8}$ | 1.50 | $1.18\times10^{-8}$ | 1.47 | $1.50\times10^{-4}$ |
| Cl | 1.51 | $8.66\times10^{-7}$ | 1.50 | $7.02\times10^{-8}$ | 1.50 | $1.18\times10^{-8}$ | 1.47 | $1.50\times10^{-4}$ |
| OC | 1.45 | 0.00 | 1.45 | 0.00 | 1.45 | 0.00 | 1.45 | 0.00 |
| BC | 1.85 | 0.71 | 1.85 | 0.71 | 1.85 | 0.71 | 1.85 | 0.71 |
| OIN | 1.55 | $3.00\times10^{-3}$ | 1.55 | $3.00\times10^{-3}$ | 1.55 | $3.00\times10^{-3}$ | 1.55 | $3.00\times10^{-3}$ |

[a] refr. index = refractive index; [b] real = real part; [c] imaginary = imaginary part

According to the source code of WRF-Chem model, the refractive index of each species was listed in Table R2. BC is a typical light-absorbing aerosol (Bond et al., 2004; 2013). Second to BC, OIN is also treated as light-absorbing aerosol since the imaginary part of which being larger than that of other species. The remaining species are treated as light-scattering aerosols. In order to showing the effects of the two types of aerosols on $J$[NO$_2$], two more parallel experiments (Exp3 and Exp4) were designed: Exp3, photolysis rate calculation without considering the optical properties of light-scattering aerosols; Exp4, photolysis rate calculation without considering the optical properties of light-absorbing aerosols. By comparing the results of Exp3 and Exp4 with the results

of Exp1 respectively, the effects of light-absorbing aerosols and light-scattering aerosols on $J$[NO$_2$] profile can be figured out.

[Figure]

Figure R3. Mean profiles of J[NO$_2$] and types of aerosols with diameter equal or less than 2.5 μg at 12:00 in clean days (a) and polluted days (b). Mean PBL height of the two kinds of days are also presented in (a) and (b), respectively.

Same as the data collection rule of Fig.3 in the manuscript but for the four experiments, the $J$[NO$_2$] profiles under the low-level aerosol condition (clean) and high-level aerosol condition (polluted) at noon (12:00) are presented in Fig. R3. Correspondingly, the profiles of the two types of aerosols (cyan and brown shades) under clean and polluted conditions are also presented in Fig. R3a and R3b, respectively. Under clean condition (Fig. R3a), aerosols were at very low levels and didn't impact $J$[NO$_2$] significantly. Consequently, the four profiles didn't show significant differences in vertical direction. Under polluted condition (Fig. R3b), the concentrations of PM$_{2.5}$ were at relatively high levels in the lowest 1.3 km (PM$_{2.5}$ with mean concentration of 90.0 μg m$^{-3}$; light-absorbing aerosols and light-scattering aerosols are 19.4 μg m$^{-3}$ and 70.6 μg m$^{-3}$, respectively), especially in the PBL, where the mean concentration of

PM$_{2.5}$ reached 123.1 μg m$^{-3}$ (light-absorbing aerosols and light-scattering aerosols are 28.4 μg m$^{-3}$ and 94.7 μg m$^{-3}$, respectively). Since light-absorbing effect of light-absorbing aerosols, the incident solar irradiance was attenuated (Ding et al., 2016; Gao et al., 2018) and $J$[NO$_2$] profile ($J$[NO$_2$]_Exp3) decreased along with the vertical direction. For light-scattering aerosols, since high concentration being located in lower layer, the incident solar radiation could be scattered backward and enhance the shortwave radiation in higher layer. In this case, $J$[NO$_2$] ($J$[NO$_2$]_Exp4) aloft was enhanced. However, the incident solar irradiance was attenuated at the layers near the surface which leading to the decrease in $J$[NO$_2$] near the surface. Combining the effects of the two types of aerosols, the light extinction of aerosols on $J$[NO$_2$] ($J$[NO$_2$]_Exp2) decreased at the lowest 1.3 km but enhanced above 1.3 km.

Unfortunately, since lacking of relevant observations of the aerosol species, concentrations of the simulated aerosols species could not be validated and this may cause some uncertainties to the impacts of different types of aerosols on $J$[NO$_2$] profiles. Thus, we just present these results and discussions in the response material. However, our validations on PM$_{2.5}$, $J$[NO$_2$], and $J$[O$_3$$^1$D] are acceptable which suggested that the results on the light extinction of aerosols on photolysis rates and its effect on ozone concentrations which we discussed in our study are meaningful. In addition, our results are consistent with that from other study (Dickerson et al., 1997) which also demonstrate the validity of the results we presented in the manuscript.

It should be noted that different contributions of aerosol species could impact photolysis rates differently. Aerosols species contributed very differently at different places. Figuring out the effects of aerosols on $J$[NO$_2$] profiles over East China is an interesting topic which being worthy of further studying.

The light extinction of aerosols can influence ozone not only via affecting photolysis rate, but also via suppressing the PBL. In this study, we mainly focus on the impact of light extinction of aerosols on photolysis rate and how this impact influence ozone. The difference between the two experiments we designed is just at whether taking optical properties of the aerosols into the calculation of photolysis rate. And other

parts of the model system are not modified. There were not significant differences in concentrations of $PM_{2.5}$ and the PBLHs between Exp1 and Exp2 (Fig. R4), especially the PBLHs from Exp1 and Exp2 are almost the same, which suggested that the effects of the collapsed PBL on ozone induced by aerosols could not be shown in this study.

[Figure]

Figure R4. Time series of the simulated $PM_{2.5}$ (a) and PBLH (b) from Exp1 and Exp2 at Xianghe station during Oct. 2018.

However, this comment is very meaningful. Effects of PBL, especially the effects of the interaction between PBL and aerosols, is an important impact on ozone which we have discussed in another paper (Gao et al., 2018). The suppression of PBL induced by the light extinction of aerosol can weakened the entrainment of turbulence aloft which leading to less ozone with high concentrations being transported down to the surface. Furthermore, the suppression of PBL can confine more ozone precursors in the PBL which may enhance the contribution of CHEM. However, when considering the decrease of photolysis rate induced by the light extinction of aerosol simultaneously, the contribution of CHEM may change differently and the change of ozone concentration may be different too. These multiple influence paths on ozone remind us that more clearly study on each impact first is very necessary. And this is also the purpose for us to conduct this study. We believe that, on the basis of this study, the effect of the interaction between PBL and aerosols on ozone will be studied more clearly in future work. And again, we really thank you for your enlightening comments.

Technical corrections

(1) References should be formatted appropriately and numbered.

Reply: Thank you for your comment. According to the format requirements of the references (https://www.atmospheric-chemistry-and-physics.net/for_authors/manuscript_preparation.html), we have reformatted all the references. However, the requirements and the format template don't list the references with numbers. Thus, numbers to references were not listed but we believe that the new reference list has become clearer than the previous version. Please check the new reference list in the revised manuscript.

(2) Please, add units to Table 2.

Reply: Thank you for your comment. Units of all the variables have been added in Table 2. Please check the new Table 2 in the revised manuscript.

(3) Line 224. Delete "which showed that ozone stopped decreasing"

Reply: Thank you very much. We follow this comment. And "which showed that ozone stopped decreasing" has been deleted. Please check the detail in the revised manuscript at lines 246~247.

(4) Figure 6, Please, update the caption, remove the CASE* and explain that data was spatially sampled and represent four cities.

Reply: Thank you for your comment. We have updated the caption of Fig. 6. Some important information has been added in it. Please check the new caption of Fig. 6 in the revised manuscript.

Reference

Bond, T. C., Streets, D. G., Yarber, K. F., Nelson, S. M., Woo, J. H., and Klimont, Z.: A technology-based global inventory of black and organic carbon emissions from

combustion, J. Geophys. Res., 109, D14203, https://doi.org/doi:10.1029/2003JD003697, 2004.

Bond, T. C., Doherty, S. J., Fahey, D. W., Forster, P. M., Berntsen, T., DeAngelo, B. J., Flanner, M. G., Ghan, S., Karcher, B., Koch, D., Kinne, S., Kondo, Y., Quinn, P. K., Sarofim, M. C., Schultz, M. G., Schulz, M., Venkataraman, C., Zhang, H., Zhang, S., Bellouin, N., Guttikunda, S. K., Hopke, P. K., Jacobson, M. Z., Kaiser, J. W., Klimont, Z., Lohmann, U., Schwarz, J. P., Shindell, D., Storlvmo, T., Warren, S. G., and Zender, C. S.: Bounding the role of black carbon in the climate system: A scientific assessment, J. Geophys. Res.: Atmos., 118, 5380–5552, https://doi.org/doi:10.1002/jgrd.50171, 2013.

Dickerson, R. R., Kondragunta, S., Stenchikov, G., Civerolo, K. L., Doddridge, B. G., and Holben, B. N.: The impact of aerosols on solar ultraviolet radiation and photochemical smog, Science, 278, 827-830, https://doi.org/10.1126/science.278.5339.827, 1997.

Ding, A. J., Huang, X., Nie, W., Sun, J. N., Kerminen, V. M., Petaja, T., Su, H., Cheng, Y. F., Yang, X. Q., Wang, M. H., Chi, X. G., Wang, J. P., Virkkula, A., Guo, W. D., Yuan, J., Wang, S. Y., Zhang, R. J., Wu, Y. F., Song, Y., Zhu, T., Zilitinkevich, S., Kulmala, M., and Fu, C. B.: Enhanced haze pollution by black carbon in megacities in China, Geophys. Res. Lett., 43, 2873-2879, https://doi.org/10.1002/2016GL067745, 2016.

Gao, J. H., Zhu, B., Xiao, H., Kang, H. Q., Pan, C., Wang, D. D., and Wang, H. L.: Effects of black carbon and boundary layer interaction on surface ozone in Nanjing, China, Atmos. Chem. Phys., 18, 7081-7094, https://doi.org/10.5194/acp-18-7081-2018, 2018.

Zaveri, R. A., Easter, R. C., Fast, J. D., and Peters, L. K.: Model for simulating aerosol interactions and chemistry (MOSAIC), J. Geophys. Res.-Atmos., 113, D13204, https://doi.org/10.1029/2007jd008782, 2008.

---

## Author Comment (AC2) · 15 Jul 2020

Dear editor and referee#1,

Thank you very much for your time and attentions on this work. The comments and suggestions are very useful to improve our manuscript. Following is a point-by-point response to referee #1's comments. Texts in black are the comments, those in blue are our responses. All the line numbers mentioned in responses are referred to the manuscript with changes marked.

We hope that you will find the changes satisfactory and we are looking forward to hearing from you soon.
* * *
(1) In model validation (section 3.1), the authors compare the simulated  $J[NO_2]$  with the observations. In addition to  $J[NO_2]$ ,  $J[O_3^1D]$  is also important in affecting the ozone photochemical production. Comparison on  $J[O_3^1D]$  will show more sufficient evidence to demonstrating the well model performance in simulating photolysis rates. If the authors have the observations of  $J[O_3^1D]$ , please add the comparison of  $J[O_3^1D]$ .

Reply: Thank you for this comment.  $J[O_3^{1}D]$  is indeed important in ozone photochemistry. Comparison on  $J[O_3^{1}D]$  is important and necessary in photolysis rates validation. However, we didn't have the data before. Fortunately, we now have gathered the observations of  $J[O_3^{1}D]$  at Xianghe station, and added the comparison of  $J[O_3^{1}D]$ in the revised manuscript. Like the comparison of  $J[NO_2]$ , both the time series of  $J[O_3^{1}D]$  and the relevant model performance metrics showed a good agreement between the observations and simulations. The model validations on  $J[NO_2]$  and  $J[O_3^{1}D]$  suggested that the WRF-Chem model performed very well in simulating the photolysis rates. Details can be checked in the revised manuscript in section 3.1.2.

(2) The authors showed that  $J[NO_2]$  was enhanced at altitude above 1.3 km which is due to the enhancement of the light caused by the light-scattering effect of aerosols.

Discussions on the compositions of the aerosols and their effects on  $J[NO_2]$  over this place are necessary. Please add them in the manuscript.

Reply: Thank you for your comment. Based on the optical properties of aerosols, they can be classified into light-scattering aerosols and light-absorbing aerosols. Before talking about the comprehensive effects of aerosols on  $J[NO_2]$ , it's necessary to present the effects of light-scattering aerosols and light-absorbing aerosols on  $J[NO_2]$ , respectively.

Figure R1. Time series (a) and mean contributions (b) of the simulated aerosol species at Xianghe station during Oct. 2018. I for the whole month; II for clean days (blue shaded parts in a); III for polluted days (yellow shaded parts in a).

In this study, MOSAIC-8bins was used as the aerosol chemistry mechanism. This mechanism includes eight aerosols species: Sulfate (SO4), Nitrate (NO3), Ammonium (NH4), Sodium (Na), Chlorine (Cl), Organic Carbon (OC), Black Carbon (BC), and, Other Inorganics (OIN). Based on Fig. 2c in manuscript, concentrations of all the simulated aerosols species and their relative contributions to the total concentration of  $PM_{2.5}$  at Xianghe station are shown in Fig. R1. During Oct. 2018, the mean concentration of  $PM_{2.5}$  was 68.0 µg m-3 at Xianghe station. Among all the species, NO3 and OIN contributed significantly which accounted for 30% and 28% to the total

concentration of PM2.5; SO4, NH4, BC, and OC accounted for ~10%, respectively; Na and Cl showed few contributions during Oct. 2018. Under the "clean" condition (blue shaded parts in Fig. R1a and the pie chart II in Fig. R1b), the mean concentration of PM2.5 decreased to 25.3  $\mu$ g m-3 and OIN contributed (accounted for 38%) more than NO3 did (accounted for 10%). On the contrary, OIN contributed (accounted for 24%) less than NO3 did (accounted for 38%) when it was under the "polluted" condition (yellow shaded parts in Fig. R1a and the pie chart III in Fig. R1b).

Table R1. Refractive indexes of the aerosol species at each wave band in WRF-Chem model

| wave band                           | 300nm             |                        | 400nm |                       | 600nm |                       | 999nm |                       |
|-------------------------------------|-------------------|------------------------|-------|-----------------------|-------|-----------------------|-------|-----------------------|
| refr. index a
species | real b | imaginary c | real  | imaginary             | real  | imaginary             | real  | imaginary             |
| SO4                                 | 1.52              | 1.00×10-9              | 1.52  | 1.00×10 -9 | 1.52  | 1.00×10-9             | 1.52  | 1.75×10-9             |
| NO3                                 | 1.50              | 0.00                   | 1.50  | 0.00                  | 1.50  | 0.00                  | 1.50  | 0.00                  |
| NH4                                 | 1.50              | 0.00                   | 1.50  | 0.00                  | 1.50  | 0.00                  | 1.50  | 0.00                  |
| Na                                  | 1.51              | 8.66×10 -7  | 1.50  | 7.02×10 -8 | 1.50  | 1.18×10 -8 | 1.47  | 1.50×10 -4 |
| Cl                                  | 1.51              | 8.66×10 -7  | 1.50  | 7.02×10 -8 | 1.50  | 1.18×10 -8 | 1.47  | 1.50×10 -4 |
| OC                                  | 1.45              | 0.00                   | 1.45  | 0.00                  | 1.45  | 0.00                  | 1.45  | 0.00                  |
| BC                                  | 1.85              | 0.71                   | 1.85  | 0.71                  | 1.85  | 0.71                  | 1.85  | 0.71                  |
| OIN                                 | 1.55              | 3.00×10 -3  | 1.55  | 3.00×10 -3 | 1.55  | 3.00×10 -3 | 1.55  | 3.00×10 -3 |

a refr. index = refractive index; b real = real part; c imaginary = imaginary part

According to the source code of WRF-Chem model, the refractive index of each species was listed in Table R1. BC is a typical light-absorbing aerosol (Bond et al., 2004; 2013). Second to BC, OIN is also treated as light-absorbing aerosol since the imaginary part of which being larger than that of other species. The remaining species are treated as light-scattering aerosols. In order to showing the effects of the two types of aerosols on  $J[NO_2]$ , two more parallel experiments (Exp3 and Exp4) were designed: Exp3,

photolysis rate calculation without considering the optical properties of light-scattering aerosols; Exp4, photolysis rate calculation without considering the optical properties of light-absorbing aerosols. By comparing the results of Exp3 and Exp4 with the results of Exp1 respectively, the effects of light-absorbing aerosols and light-scattering aerosols on *J*[NO2] profile can be figured out.

---

## Author Response (AR2)

Dear editor,

Thank you very much for your comment. Relevant discussions and figures in the response to referees have been added to the supplementary material, and reference them from the main text. Please check the details in the new supplementary material and revised manuscript (lines 204 and 205).

We hope that you will find the changes satisfactory and we are looking forward to hearing from you soon.

[revised manuscript text omitted]